# Effects of Conservation Tillage on Soil Properties and Maize Yield in Karst Regions, Southwest China

**Lizhen Bai [1], Xiangying Kong [2], Hui Li [3], Huibin Zhu [1,*], Chengwu Wang [1] and Shiao Ma [1]**

[1] Faculty of Modern Agricultural Engineering, Kunming University of Science and Technology, Kunming 650500, China
[2] Faculty of Architecture and City Planning, Kunming University of Science and Technology, Kunming 650500, China
[3] Shandong Academy of Agricultural Machinery Sciences, Ji'nan 250100, China
* Correspondence: hbzhu113@163.com or hbzhu113@kust.edu.cn; Tel.: +86-13330490820

**Abstract:** Karst rocky desertification associated with human disturbance is one of the most serious eco-environmental problems, threatening the sustainable development of agriculture in southwestern China. In the current study, the practice of conservation tillage as one of the best ways of reducing the constraints is addressed. During a two-year trial (2014–2015), the effects of no tillage with straw cover (NT) and traditional tillage (TT) on soil properties and maize yields were investigated in karst regions, Southwest China. The results showed that the trial with NT increased soil moisture content by 3%, while decreasing soil bulk density by 7% in the top 30 cm compared with TT. In 2014, within 0–30 cm of soil depth, total nitrogen under NT treatment was 5% higher than that under TT treatment. In 2015, the mean soil organic matter (SOM) and available P were enhanced to 12% and 13% in 0–30 cm soil depth more than that under TT, respectively. The trial with NT significantly ($p < 0.05$) increased available N in the top 20 cm by 9% as compared to TT. This improvement in soil physical and chemical properties might have increased the crop yield. After the two-year trial with NT, the mean maize yields increased by 11% compared with the TT trial. Therefore, conservation tillage is the better option considering long-term environmental sustainability in karst regions.

**Keywords:** karst rocky desertification; conservation tillage; no tillage; soil properties; maize yield; traditional tillage

## 1. Introduction

Southwest China (including Guizhou, Yunnan, Sichuan etc.) has about 45 Mha of karst rocky desertification, which not only causes serious ecological problems, but is also an important factor restricting regional economic and social development. The karst landscape degeneration, which has been caused by human activity, challenges the restoration and stability of the limestone ecosystem [1]. Therefore, more attention is required from both the government and the public [2,3]. In southwestern China, karst rocky desertification was caused by irrational land use on the fragile, thin karst soil in the karst mountains [4,5], and, most importantly, intensive artificial activities.

Conservation tillage is defined as any tillage and planting system that leaves 30% of crop residue on the soil surface after planting [6]. No tillage, shallow surface tillage, subsoiling, strip rototilling and residue mulching are often included under the umbrella of this definition [7,8]. Singh et al. [9] have shown that management practices based on conservation agriculture (CA), such as dry direct seeding rice (DSR), zero tillage (ZT), and residue retention, may contribute to improving yields, reducing costs and increasing farmers' profits in the rice-maize system (RMS). Shahzad et al. [10] showed that the interaction between different tillage methods and tillage systems had significant effects on soil bulk weight and total porosity, allometric growth and yield of wheat. Su et al. [11] created a data set that helps to gain a deeper understanding of the major drivers of NT productivity

variability and its impact on crop yields. Parihar et al. [12] found that tillage and rotation had significant effects on soil organic carbon, physical properties and enzyme activity ($p < 0.05$). Singh et al. [13] emphasized the importance of adequate potassium (K) nutrition for continuous RMS enhancement. Their research found a positive correlation between exchangeable potassium in soil and potassium input, indicating that soil potassium mining in RMS can be largely alleviated by stubble retention and adequate supply of potassium fertilizer. Chen et al. [14] have shown that the small mechanical controlled transportation system on the Loess Plateau is an energy-saving system with high total energy output. Latsch et al. [15] studied the effects of permanent driveway on soil permeability resistance, water infiltration rate, bulk weight, macropore volume and yield. Numerous researchers have demonstrated that conservation tillage is effective in improving crop yields, soil physical and chemical properties, and in reducing energy consumption and production costs. In China, several long-term experiments [16,17] have generally confirmed the improvements in soil quality and productivity achieved by conservation tillage in dryland farming areas in North China. However, little is known about the effects of conservation tillage practices on soil properties and yields in karst regions, especially those in Southwest China.

A two-year comparison experiment, including no tillage with straw cover (NT) and traditional tillage with conventional ploughing (TT), was conducted in this study. The objective was to improve our understanding of the effects of conservation tillage in karst regions, and, particularly, to quantify an assessment of the potential benefits on soil properties and crop yields.

## 2. Materials and Methods

### 2.1. Site Description

The experiment was conducted from 2014 to 2015 in Yiliang county (24°30′–25°17′ N, 102°58′–103°28′ E, 1600 m a.s.l., Figure 1), which is in a moderate rocky desertification area, and is situated in middle Yunnan province, Southwest China. The experimental area has a subtropical plateau monsoon climate. The average annual temperature is 16.3 °C with 260 frost-free days. The average annual rainfall is 912 mm, and more than 67% of the rainfall occurs during June–October. The main type of soil in Yiliang is red soil. Before the experiment, the condition of the soil chemical and physical property conditions at the site were assessed and are reported in Table 1.

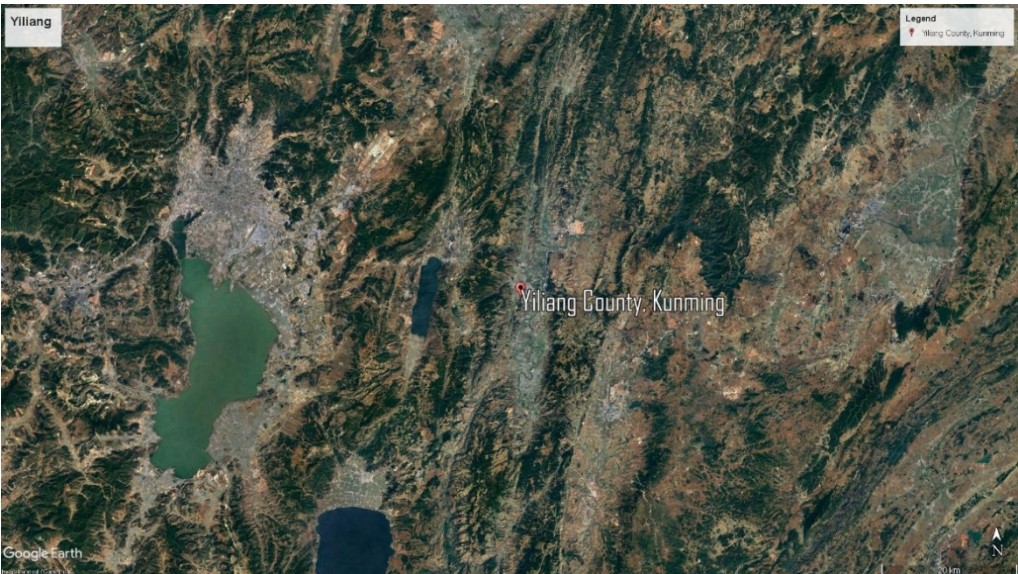

**Figure 1.** Location map of the study area.

**Table 1.** Basic physical and chemical properties of the soil in November 2013.

| Soil Depth (cm) | Bulk Density (g·cm$^{-3}$) | SOM (g·kg$^{-1}$) | Total N (g·kg$^{-1}$) | Available N (mg·kg$^{-1}$) | Available P (mg·kg$^{-1}$) |
|---|---|---|---|---|---|
| 0–10 | 1.19 | 25.32 | 1.16 | 85.99 | 8.88 |
| 10–20 | 1.27 | 23.28 | 0.94 | 75.71 | 7.84 |
| 20–30 | 1.36 | 20.06 | 0.88 | 63.15 | 5.72 |

*2.2. Experimental Design*

The experiment was designed as a randomized block with five replications. Each plot was 4 m wide and 15 m long. Two tillage/residue treatments were used: no tillage with straw cover (NT) and traditional tillage with conventional ploughing (TT). NT normally consisted of no-till seeding (to 5 cm depth) and fertilizing through the previous plant residues with Lee-seeder I. TT included spreading of fertilizer, mold board ploughing (to 15 cm depth) and seeding (to 5 cm depth).

During the two-year trial (2014–2015), seed and fertilizer were commonly applied at very high rates by farmers to maximize the chance of good yields. The summer maize ('Xindan 4') was sown from June 10 to 20 every year with a density of $6.66 \times 10^4$ ha$^{-1}$ and harvested from October 9 to 15. Basal fertilizer was applied at the rate of 120 kg N·ha$^{-1}$, 120 kg P$_2$O$_5$·ha$^{-1}$, and 100 kg K$_2$O·ha$^{-1}$. An amount of 100 kg N was applied as topdressing fertilizer at the maize jointing stage. Roundup (glyphosate, 10%) was used for weed control during maize growing season. Planting specifications and field management were the same as those of general high-yielding fields.

*2.3. Measured Parameters*

2.3.1. Bulk Density and Soil Moisture

After maize harvest every year, soil samples of different depths (0–10 cm, 10–20 cm, 20–30 cm) were collected. In three repeated plots, five soil samples were randomly taken using a 54-mm diameter steel core sampling tube, manually driven into a 30 cm depth. These moisture samples were then weighed wet, dried at 105 °C for 48 h, and weighed again to determine bulk density and gravimetric soil moisture [18]. Gravimetric water content was multiplied by soil bulk density to obtain volumetric water content.

2.3.2. Soil Sampling and Preparation

In October 2014, soil samples were collected after maize harvest. In each plot, one composite soil sample formed by five sub-samples was obtained at 0–10 cm, 10–20 cm and 20–30 cm soil depths. These samples were to determine soil organic matter (SOM), total N, available N, and available P. Each soil sample was first passed through an 8 mm sieve by gently breaking apart the soil and then air drying for further processing.

2.3.3. SOM, Total N, Available N and Olsen's P

Soil organic matter (SOM) was measured by dry combustion using a Leco Carbon Analyzer [19]. Total nitrogen concentration was determined by Kjeldahl digestion. Available Nitrate was extracted with 1 M KCl and analyzed by the cadmium reduction method [20]. Available potassium was extracted with 1 M ammonium acetate and analyzed by flame photometer. Available potassium (Olsen's phosphorus) was extracted with 0.5 M NaHCO$_3$ solution adjusted to pH 8.5 [21]. All the measurements were replicated 3 times.

2.3.4. Yield

In the field experiment, 10 m three row sampling was used and the output area was 18 m$^2$. Harvesting of the maize ears, drying, and measuring of yield were repeated three times. The number of rows per ear were multiplied by the average number of five randomly chosen rows of maize grain to obtain the number of grains per ear. After the ears of maize

were dried, the grains were mechanically separated from the ears, and then dried at 65 °C for 48 h to measure the dry weight.

### 2.4. Statistical Analysis

Mean values were calculated for each of the variables, and ANOVA was used to assess the effects of NT and TT on the measured soil parameters and crop yields. Significance of the F-value was determined from ANOVA tables. Multiple comparisons of annual mean values were performed by the least-significant-difference method (l.s.d.). The SPSS (SPSS for Windows, version 20.0, Armonk, NY, USA), Origin 2017 (OriginLab Corporation, Northampton, MA, USA.) and Microsoft Excel 2016 (Microsoft Corporation, Redmond, WA, USA) were used for data processing and statistical analysis. In all analyses, a probability of error smaller than 5% ($p < 0.05$) was considered marked.

## 3. Results

### 3.1. Bulk Density and Soil Moisture

#### 3.1.1. Bulk Density

The mean soil bulk density for NT and TT treatments in 0–30 cm soil depth was 1.27 g·cm$^{-3}$ in the experimental site in November, 2013. After the two-year trial of different tillage, soil bulk density in the no tillage with straw cover (NT) plots declined slightly, compared with traditional tillage (TT) (Figure 2). In 20–30 cm soil layers, the mean bulk density in NT treatment was 5.6% ($p < 0.05$), significantly lower than in TT in 2014. At the same depths, the mean bulk density in NT treatment was 11.6% ($p < 0.05$), notably lower than in TT in 2015. A similar trend was observed at the 20–30 cm soil depth, but the difference between NT and TT was negligible.

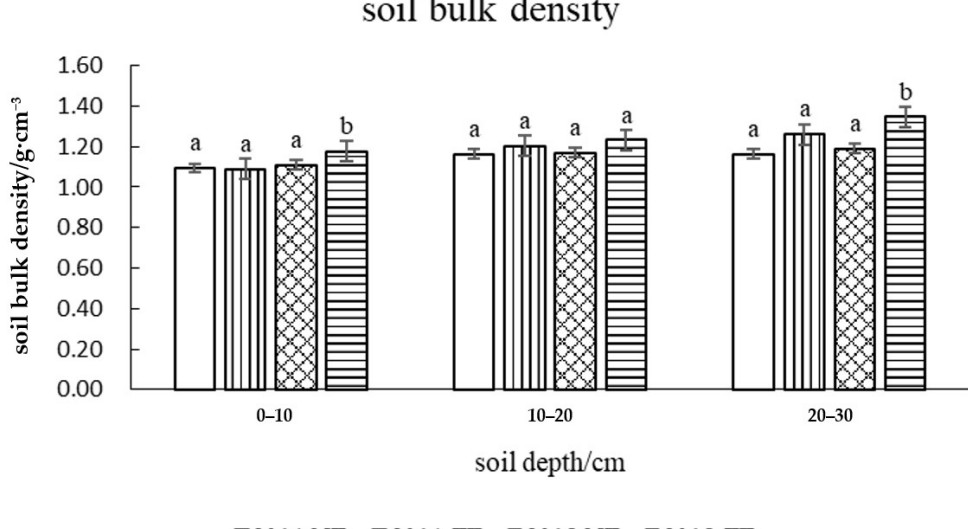

**Figure 2.** Mean bulk density of NT and TT treatments in 0–10 cm, 10–20 and 20–30 cm depths in 2014 and 2015. Samples were taken immediately after maize harvest in October every year. Means within each depth followed by the same letter were not significantly different ($p > 0.05$); NT: no tillage with straw cover; TT: traditional tillage with conventional ploughing.

In 2019, the measured soil bulk density of the 0–5 cm soil layer was 0.98 g·cm$^{-3}$, of the 5–10 cm soil layer it was 1.06 g·cm$^{-3}$, of the 10–15 cm soil layer it was 1.12 g·cm$^{-3}$, of the 15–20 cm soil layer it was 1.10 g·cm$^{-3}$, of the 20–25 cm soil layer it was 1.26 g·cm$^{-3}$ and of the 25–30 cm soil layer it was 1.23 g·cm$^{-3}$.

According to 2019 data, the soil bulk density was 1.14 g·cm$^{-3}$ in the 0–30 cm soil layer. That was 11.40% lower than in 2013. Compared with the data in 2015, the soil bulk weight

decreased by 6.80% in the 0–10 cm soil layer, and by 4.05% in the 10–20 cm soil layer, and remained basically unchanged in the 20–30 cm soil layer.

### 3.1.2. Soil Moisture

After the two-year trial of different tillage, soil moisture content in the no tillage with straw cover (NT) plots improved, compared with traditional tillage (TT) (Figure 3). In 20–30 cm soil layers, the mean soil moisture content in NT treatment was 30% ($p < 0.01$), markedly higher than in TT in 2015. A similar trend was observed at the 0–10 cm and 10–20 cm soil depth, but the difference between NT and TT was negligible.

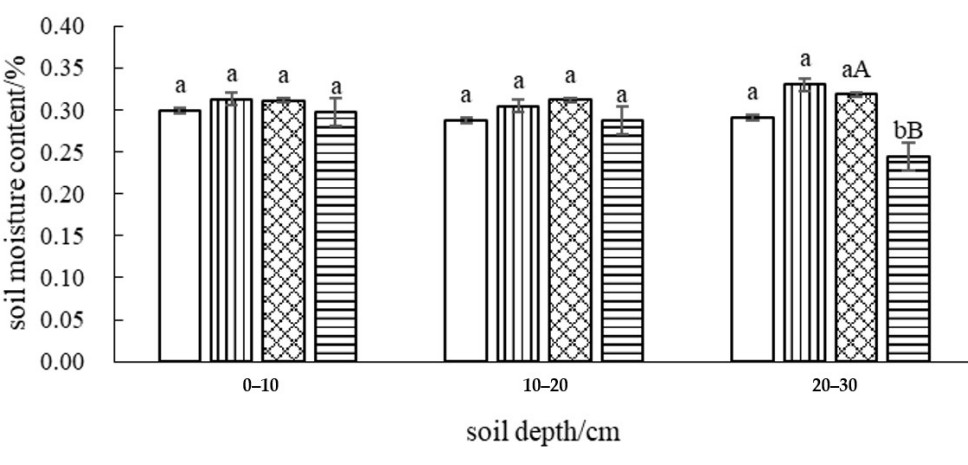

**Figure 3.** Mean soil moisture content of NT and TT treatments in 0–10 cm, 10–20 and 20–30 cm depths in 2014 and 2015. Samples were taken immediately after maize harvest in October every year. Means within each depth followed by the same letter were not significantly different ($p > 0.05$); NT: no tillage with straw cover; TT: traditional tillage with conventional ploughing.

In 2019, the water content of the 0–5 cm soil layer was 23.37%, the water content of the 5–10 cm soil layer was 24.27%, the water content of the 10–15 cm soil layer was 26.24%, the water content of the 15–20 cm soil layer was 28.74%, the water content of the 20–25 cm soil layer was 29.38%, and the water content of the 25–30 cm soil layer was 30.68%.

Compared with the data in 2015, the soil water content decreased by 28.98% in the 0–10 cm soil layer, by 12.48% in the 10–20 cm soil layer, and by 5.05% in the 20–30 cm soil layer. In 2019, soil water content showed a significant downward trend. This was due to severe Spring-Summer Drought in Yunnan Province in 2019 [22]. From April to June, the average precipitation in Yunnan Province decreased by 42.9%, the lowest since 1961. The average temperature in the whole province was higher than 1.9 °C, which was the highest in the same period in history.

### 3.2. SOM, Total N, Available N, and Available P

Soil organic matter, total N, available N, and available P results are presented in Table 2. In the 0–10 cm and 10–20 cm soil depths, SOM, total N, available N, and available P were higher than at deeper (20–30 cm) soil depth. For the different treatments, at the beginning of the experiment in 2014, the differences between NT and TT were not critical, but pronounced treatment effects on SOM, available N, and available P could be observed after the two-year trial.

**Table 2.** SOM (g·kg$^{-1}$), total N (g·kg$^{-1}$), available N (mg·kg$^{-1}$), and available P (mg·kg$^{-1}$) for NT and TT trial in 0–30 cm depth.

| Year | Depth/cm | SOM (g·kg$^{-1}$) | | Total N (g·kg$^{-1}$) | | Available N (mg·kg$^{-1}$) | | Available P (mg·kg$^{-1}$) | |
|---|---|---|---|---|---|---|---|---|---|
| | | NT | TT | NT | TT | NT | TT | NT | TT |
| 2014 | | | | | | | | | |
| | 0–10 | 27.13(a) | 25.48(a) | 1.21(a) | 1.19(a) | 89.7(a) | 83.76(a) | 9.22(aA) | 8.77(bB) |
| | 10–20 | 24.85(a) | 23.87(a) | 1.04(a) | 0.93(b) | 77.36(a) | 73.53(a) | 8.08(a) | 7.83(b) |
| | 20–30 | 20.87(a) | 19.83(a) | 0.95(a) | 0.89(a) | 65.86(a) | 65.81(a) | 6.27(aA) | 5.95(bB) |
| 2015 | | | | | | | | | |
| | 0–10 | 28.55(aA) | 24.28(bB) | 1.27(a) | 1.14(b) | 91.97(aA) | 84.65(bB) | 10.56(aA) | 8.95(bB) |
| | 10–20 | 25.73(a) | 23.15(b) | 1.06(a) | 0.93(b) | 79.77(a) | 73.40(b) | 8.91(aA) | 8.06(bB) |
| | 20–30 | 21.41(a) | 19.79(a) | 0.96(a) | 0.84(b) | 67.35(a) | 61.25(a) | 6.32(a) | 5.84(b) |

Note: NT: no tillage with straw cover; TT: traditional tillage with conventional ploughing. The data were tested after maize harvest in October every year. Different letters indicate significant differences between different tillage measure: values within the same column, data followed by the same letters are not significantly different using LSD. Lowercase letters, uppercase letters significant at $p \leq 0.05$ and $p \leq 0.01$, respectively.

The mean of SOM to 0–30 cm soil depth for NT in 2015 was 0.95 g·kg$^{-1}$ greater than that in 2014, while TT reduced SOM by 0.59 g·kg$^{-1}$ in 2015 relative to 2014. Consequently, SOM in 0–30 cm for NT was approximately 4% higher than for TT after the two-year trial. In the surface soil layer (0–10 cm), the mean SOM was 28.55 g·kg$^{-1}$ for the no-till plot, which was enormously ($p < 0.01$) greater than the 24.48 g·kg$^{-1}$ observed on the TT plot. SOM in 10–20 cm for NT was 2.58 g·kg$^{-1}$ greater than that of TT in 2015. In the deeper (20–30 cm) soil layer, however, inconsequential differences were observed between NT and TT treatments.

Soil total N showed the same trend as SOM in relation to tillage treatments. Mean of total N in 0–30 cm soil depth improved by 3% on NT while it reduced by 3% on TT. Mean of total N to 0–30 cm soil depth for NT was 1.10 mg·kg$^{-1}$, and approximately 13% higher than that for TT in 215.

Compared to 2014, mean available N in 0–30 cm soil depth improved by 3% on NT while it reduced by 2% on TT during the two-year trial. In the 0–10 cm soil depth, available N under NT soils was 9% ($p < 0.01$), which was significantly higher than that under TT soils, while in the 10–20 cm soil layers, available N under NT soil was 6.37 mg·kg$^{-1}$ greater than that under TT soils. In the deeper (20–30 cm) soil layer, the available N differences were insignificant.

The quantity of available P in both treatments was very similar, but notable differences developed across the soil profile during the two-year trial. In 2015, the available P under NT was 15% higher than under TT in the 0–10 cm soil layer, impressive at $p < 0.01$, while in the 10–20 cm layer, the available P content was 10% statistically ($p < 0.01$) greater under NT than under TT. In the 20–30 cm layer the difference was statistically 8% ($p < 0.05$).

*3.3. Yield*

Maize yield in NT and TT treatments fluctuated slightly from 2014 to 2015 (Figure 4). Mean maize yield for no tillage with straw cover (NT) was about 11% greater than that for traditional tillage (TT) during the two-year trial. Mean maize yield for NT in 2015 was 8.6% greater than that in 2014, while TT increased by 8.2% in 2015 relative to 2014. It was observed that the mean yield advantage of NT was not significantly different ($p > 0.05$).

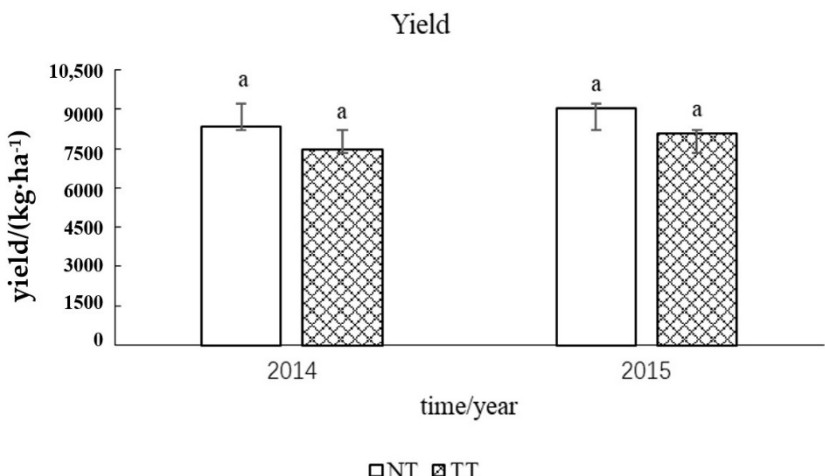

**Figure 4.** Summer maize yield from 2014 to 2015; NT: no tillage with straw cover; TT: traditional tillage with conventional ploughing. Means within each depth followed by the same letter were not significantly different ($p > 0.05$).

There was no significant change in maize production in 2019 compared to 2015. The main reason being the drought in Yunnan Province in 2019.

## 4. Discussion

The results of the two-year trial clearly demonstrated that no tillage with straw cover (NT) was associated with a substantial and significant improvement in soil properties and nutrient status in the karst regions of Southwest China compared to traditional tillage (TT). All relevant soil properties (SOM, total N, available N and available P content, bulk density, soil moisture) were improved and led to higher yields. Overall, the benefits of NT were greater than for TT.

In our study, the vastly higher SOM of the surface soil layer in NT treatment was attributed to greater carbon input from residue retention and reduced biological oxidation of soil organic C to $CO_2$ [22]. On the other hand, frequent and excessive tillage in TT resulted in significant SOM loss. Tillage induced changes in soil organic N are often directly related to changes in soil organic C [23]. NT had remarkably ($p < 0.05$) greater total N in the soil layer (0–30 cm). Soil available N in the surface soil layer (0–10 cm) was significant ($p < 0.01$), while the deeper layer (20–30 cm) was not affected. Soil available P in 0–10 cm depth also increased under NT, confirming the finding of Zhang et al. [24]. The topsoil accumulation of available P in NT could be explained by the limited downward movement of particle bound P in NT soils and the upward movement of nutrients from deeper layers through nutrient uptake by roots [25]. The lack of soil turnover also explained the lower amount of available P in NT treatment below 20 cm depth compared to TT. In the semi-arid region of the Indo-Gangetic Plains, Singh A., et al. [26] reported that ZT (zero tillage, direct drilling) increased soil organic carbon significantly to a depth of 0.10, 0.15 and 0.25 m in sandy loam, loam and clay loam soil, respectively, indicating its buildup to deeper depths with increase in fineness of soil texture. The significant increases of available N and available P in NT were also consistent with the findings of other researchers [27,28].

Frequent and excessive ploughing in TT led to soil compaction and the formation of a plough pan in the lower soil profile [29]. In our study, at the beginning of the experiment in 2014, the bulk density differences between NT and TT treatments were slight. However, during the two-year trial, the mean bulk density in the top 30 cm on NT plots was enormously lower than on the TT plots ($p < 0.05$), especially below the ploughing depth of the TT treatment (Figure 4). These results suggested that the increased soil bulk density of the early years on NT plots was balanced over time by other changes in the soil, for example the greater amount of soil organic C and greater aggregate stability [30]. The results of Li H. et al. [31] also showed that the overall soil bulk density (0–30 cm) in PRB (permanent

raised beds) plots was markedly ($p < 0.05$) lower (by 12.4%) than that in traditional tillage (TT) plots in Beijing. Our data were consistent with those.

Soil moisture increased as a consequence of the improved soil properties. During the two-year trial, the mean soil moisture content in the 0–30 cm soil profile of NT was 3% greater than on TT, demonstrating that ploughing resulted in less soil water retention capacity and higher soil moisture loss compared to NT. These results might be explained by the greater surface area for evaporation and greater gas permeability after ploughing.

It was interesting to note that the yield increased due to NT (Figure 4). The mean yields of summer maize for NT were about 11% greater than for TT treatment. The positive effects of NT and residue cover on grain yields were consistent with other reported results [32–34]. Compared to TT, NT improved grain yields about 10% in similar climatic conditions.

## 5. Conclusions

In karst regions of Southwest China, no tillage with straw cover (NT) is advocated as a sustainable agricultural option. A two-year trial with NT led to significant positive effects on soil properties and maize yields. We observed increased soil organic matter (22.4 g·kg$^{-1}$–25.2 g·kg$^{-1}$), total N (0.97 g·kg$^{-1}$–1.09 g·kg$^{-1}$), available N (73.1 mg·kg$^{-1}$–79.7 mg·kg$^{-1}$), available P (7.6 mg·kg$^{-1}$–8.6 mg·kg$^{-1}$), and soil moisture content (29–32%), and decreased soil bulk density (1.27 g·cm$^{-3}$–1.16 g·cm$^{-3}$), compared to conventional tillage-based maize production. This considerably increased maize yield for the NT trial, which improved by 11% compared to the TT trial.

In general, the results suggested an incipient differentiation of soil properties after two years of conservation tillage, already linked to an increased maize yield. Our study suggests that NT provides an alternative to improving soil properties in karst soil of Southwest China and other similar karst regions.

**Author Contributions:** Conceptualization, L.B. and H.Z.; Formal analysis, L.B., X.K. and H.Z.; Funding acquisition, H.Z.; Methodology, L.B. and X.K.; Supervision, X.K. and H.L.; Validation, H.L., C.W. and S.M.; Writing—original draft, L.B. and H.Z.; Writing—review & editing, C.W. and S.M. All authors have read and agreed to the published version of the manuscript.

**Funding:** This study is very grateful for the supported of the National Natural Science Foundation of China (Grant No. 51865022) and General Project of Yunnan Provincial Department of science and technology (Grant No. 2015FB125). The authors thank all the people, who provided their input to this work.

**Institutional Review Board Statement:** Not applicable.

**Informed Consent Statement:** Not applicable.

**Data Availability Statement:** The analyzed datasets are available from the corresponding author on reasonable request.

**Acknowledgments:** The authors acknowledge the support by the National Natural Science Foundation of China (Grant NO. 51865022).

**Conflicts of Interest:** The authors declare no conflict of interest.

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
