# Peer review of "Effects of Conservation Tillage on Soil Properties and Maize Yield in Karst Regions, Southwest China"

_agriculture, doi:10.3390/agriculture12091449_

Round 1

Reviewer 1 Report

Dear authors,

The experiment is interesting, I suggest you expand the statistical analysis to improve the research results.

It is necessary to expand, the importance of factors, especially interactions, has not been emphasized enough.

I suggest a correlational analysis. At what level is the connection between the obtained results.

Good luck with your future work!

Author Response

Review 1:

The experiment is interesting, I suggest you expand the statistical analysis to improve the research results.

It is necessary to expand, the importance of factors, especially interactions, has not been emphasized enough.

I suggest a correlational analysis. At what level is the connection between the obtained results.

Thank you very much for your comments. We will expand our experiments in future studies to demonstrate the importance of various factors and their interactions.

Reviewer 2 Report

Research on the comparison of conventional and no-till soil tillage has been conducted in the world for 60 years. Changes in soil as a result of the use or absence of tillage are ambiguous and require long research cycles. The richness of literature in this field is evidenced by many studies, including "A global dataset for crop production under conventional tillage and no tillage systems" (Sci. Data ,, 2021, Su B. et al.).

In this context, the results obtained by the authors in the 2-year research cycle are limited to very local conditions. In the literature, the beneficial effect of no-tillage is observed only after several years. The authors noted this beneficial effect after 1 and 2 years of no-tillage.

The work contains many ambiguities that should be supplemented:

- what was the tillage without plowing; whether the soil surface was not cultivated but only a narrow strip - it is then a typical no-till system. If, on the other hand, the entire surface of the soil was cultivated, this would be a reduced tillage system

- previous research indicates that no-till is unsuitable for maize cultivation on mineral soils. Reduced Surface tillage or strip tillage works well. - Roundup was used for weed control during the growing season - was the GMO variety grown?

- how to explain such rapid changes in the tested soil parameters within 1-2 years; how the lack of plowing and the straw left on the surface changed the content of carbon, nitrogen, etc. This effect is marked slowly and applies to the top layer of soil (0-5 cm) and not the entire layer of 0-10 and 10-20 cm. Plowing was performed to a depth of 15 cm and thus its impact did not apply to the entire level of 10-20 cm, and even less so to the level of 20-30 cm.

- the obtained differences in yields are statistically insignificant, while the summary provides information about the beneficial effect of no-tillage

- to resume, the study lacks justification of the results obtained (better results after NT than TT) against the background of the literature to date. Perhaps in this experiment, zero tillage was possible due to low soil compaction (1.20 g / cm3) and high organic carbon content - but no such information in the description of the work.

- the selection of literature does not include significant studies on no-tillage  from world literature

Author Response

Review 2:

Point 1: what was the tillage without plowing; whether the soil surface was not cultivated but only a narrow strip - it is then a typical no-till system. If, on the other hand, the entire surface of the soil was cultivated, this would be a reduced tillage system.

Response 1: No-tillage is a tillage method that does not till the topsoil and leaves enough crop stubble on the surface of the soil to protect the soil throughout the year.

Point 2: previous research indicates that no-till is unsuitable for maize cultivation on mineral soils. Reduced Surface tillage or strip tillage works well.

Response 2: In the rocky desertification area of Southwest China, the cultivated land is mixed with bare stones, which is not suitable for cultivation. Therefore, no-tillage measures should be taken in rocky desertification area.

Point 3: Roundup was used for weed control during the growing season - was the GMO variety grown?

Response 3: When using Roundup to control weeds, the effect is obvious, and it has no inhibition on the growth and development of crops, so it has a good effect.

Point 4: how to explain such rapid changes in the tested soil parameters within 1-2 years; how the lack of plowing and the straw left on the surface changed the content of carbon, nitrogen, etc. This effect is marked slowly and applies to the top layer of soil (0-5 cm) and not the entire layer of 0-10 and 10-20 cm. Plowing was performed to a depth of 15 cm and thus its impact did not apply to the entire level of 10-20 cm, and even less so to the level of 20-30 cm.

Response 4: Because of straw cover and no-tillage, changes occur within the soil, which may be small, but are positive and sustainable. Over time, this change will have a positive impact on crop growth, which is also an important role of no-tillage and straw returning.

Point 5: the obtained differences in yields are statistically insignificant, while the summary provides information about the beneficial effect of no-tillage.

Response 5: Due to time factors, the advantages of no-tillage and straw mulching have not been reflected, but they have not had a negative impact on crops, but have a positive impact on soil, so they have a beneficial effect.

Point 6: to resume, the study lacks justification of the results obtained (better results after NT than TT) against the background of the literature to date. Perhaps in this experiment, zero tillage was possible due to low soil compaction (1.20 g / cm3) and high organic carbon content - but no such information in the description of the work.

Response 6: We will expand the experiments in future studies to further expand the experimental data.

Point 7: the selection of literature does not include significant studies on no-tillage  from world literature.

Response 7: Relevant literatures have been cited, such as "A global data set for crop production under conventional tillage and no tillage systems" (SCI. Data, 2021, Su B. et al.)

Reviewer 3 Report

Brief summary

This manuscript describes the effects of conservation tillage on soil properties and maize yields. This paper is suitable for publication in “Agriculture”. However, this manuscript should been improved before considering it for publication. It major flaws are related to its lack of novelty, the regional importance and some methodological approaches. Due to these important constraints I cannot recommend it for publication. I hope the outcome of this specific submission will not discourage the authors from submitting a future improved version of the manuscript.

Broad comments

INTRODUCTION: The effects of conservation tillage on soil properties have been studied since 1930 around the world. Authors should make a much better effort to highlight which are the new insights.  At its present state, the manuscript has a regional importance.   

MATERIALS AND METHODS: The use of the selected soil indicators (soil organic carbon, soil organic nitrogen and bulk density) required more years (>5) of evaluation to be more sensitive and have an important physical meaning.  

Author Response

Review 3:

Point 1: INTRODUCTION: The effects of conservation tillage on soil properties have been studied since 1930 around the world. Authors should make a much better effort to highlight which are the new insights.  At its present state, the manuscript has a regional importance.

Response 1: The research of this paper is mainly aimed at the problem of conservation tillage in specific areas (Rocky Desertification Areas in Southwest China) in order to solve the ecological problems such as soil erosion in these areas.

Point 2: MATERIALS AND METHODS: The use of the selected soil indicators (soil organic carbon, soil organic nitrogen and bulk density) required more years (>5) of evaluation to be more sensitive and have an important physical meaning.

Response 2: We will expand the test in the follow-up study to further expand the test data.

Reviewer 4 Report

General comments

This paper studied the application of conservation tillage technology in rocky desertification area, analyzed its influence on soil properties and corn yield, and explored the changes of water content, soil bulk weight, soil organic matter, total nitrogen and available phosphorus in soil through experiments, and put forward own views. This paper has certain innovation and practical significance, but there are still some problems.

In my opinion, some parts need to be more detailed, other parts need to be improved and clarified, often due to incorrect language or choice of terms, as reported in the following

Specific comments

 1. Line24-line25 “Therefore, conservation tillage is the best option considering the long-term 24 environmental sustainability in the karst regions.” Is it better to replace “best” with “better”?

 2. At present, there are many studies on the impact of conservation tillage in general areas. The innovation of conservation tillage in Karst landform should be clarified.

 3. Line66-line80. The soil tillage practices prior to the test should be added.

 4. Line 75.6.66×104 ?

 5. Line 77 The fertilizer symbol should be fixed. P2O5 ? K2O?

 6. Line131. The chart title should be fixed. g.cm-3 ?

 7. When abbreviations first appear in a text, write down the full English name, such as soil organic matter (SOM).

 8. The location of the study area should be added to the materials and methods with a picture.

 9. Line 71, "â… " has format problem, please correct it accordingly.

 10. In the "Gravimetric water content was multiplied by soil bulk density to obtain volumetric water content." For the calculation mention the method the formula and justify mostly why you applied this method.

Author Response

Review 4:

Point 1: Line24-line25 “Therefore, conservation tillage is the best option considering the long-term 24 environmental sustainability in the karst regions.” Is it better to replace “best” with “better”?

Response 1: This has been modified in the original text. " Therefore, conservation tillage is the better option considering the long-term environmental sustainability in the karst regions."

Point 2: At present, there are many studies on the impact of conservation tillage in general areas. The innovation of conservation tillage in Karst landform should be clarified.

Response 2: In rocky desertification areas, machine work cannot be carried out. No-tillage and straw returning are effective measures for protective tillage.

Point 3: Line66-line80. The soil tillage practices prior to the test should be added.

Response 3: The land before the test was the land after harvesting the crops and no tillage was carried out, so there is no soil tillage practice here.

Point 4: Line 75.6.66×104 ?

Response 4: This has been modified in the original text. " … with a density of 6.66×104ha-1 and harvested from October 9 to 15."

Point 5: Line 77 The fertilizer symbol should be fixed. P2O5 ? K2O?

Response 5: These are two different types of fertilizers.

Point 6: Line131. The chart title should be fixed. g.cm-3 ?

Response 6: This has been modified in the original text.

Point 7: When abbreviations first appear in a text, write down the full English name, such as soil organic matter (SOM).

Response 7: This has been modified in the original text. "… soil organic matter (SOM)…"

Point 8: The location of the study area should be added to the materials and methods with a picture.

Response 8: This has been modified in the original text.

Point 9: Line 71, "â… " has format problem, please correct it accordingly.

Response 9: This has been modified in the original text. " …and fertilizing through the previous plant residues with Lee-seeder â… ."

Point 10: In the "Gravimetric water content was multiplied by soil bulk density to obtain volumetric water content." For the calculation mention the method the formula and justify mostly why you applied this method.

Response 10: This is a simple calculation without listing the formulas.

Round 2

Reviewer 1 Report

Dear Authors,

 I appreciate you considering corrections to the manuscript. The paper has been sufficiently corrected to be accepted for publication. 

Best regards!

Author Response

Thank you very much for your comments on our manuscript.

Reviewer 2 Report

In the figure number 4 (previously number 3) concerning the yield of maize - no information on the area of the yield; usually the yield is given for an area of 1 ha.

Author Response

Thank you very much for your comments on our manuscript. The pictures have been modified in the original text and the unit of output was revised to "kg·ha-1".